# Comparative transcriptomics of Venus flytrap (*Dionaea muscipula*) across stages of prey capture and digestion

**Jeremy D. Rentsch**[1]*, **Summer Rose Blanco**[2], **James H. Leebens-Mack**[2]

1 Department of Biology, Francis Marion University, Florence, SC, United States of America, 2 Department of Plant Biology, University of Georgia, Athens, GA, United States of America

* jrentsch@fmarion.edu

## Abstract

The Venus flytrap, *Dionaea muscipula*, is perhaps the world's best-known botanical carnivore. The act of prey capture and digestion along with its rapidly closing, charismatic traps make this species a compelling model for studying the evolution and fundamental biology of carnivorous plants. There is a growing body of research on the genome, transcriptome, and digestome of *Dionaea muscipula*, but surprisingly limited information on changes in trap transcript abundance over time since feeding. Here we present the results of a comparative transcriptomics project exploring the transcriptomic changes across seven timepoints in a 72-hour time series of prey digestion and three timepoints directly comparing triggered traps with and without prey items. We document a dynamic response to prey capture including changes in abundance of transcripts with Gene Ontology (GO) annotations related to digestion and nutrient uptake. Comparisons of traps with and without prey documented 174 significantly differentially expressed genes at 1 hour after triggering and 151 genes with significantly different abundances at 24 hours. Approximately 50% of annotated protein-coding genes in Venus flytrap genome exhibit change (10041 of 21135) in transcript abundance following prey capture. Whereas peak abundance for most of these genes was observed within 3 hours, an expression cluster of 3009 genes exhibited continuously increasing abundance over the 72-hour sampling period, and transcript for these genes with GO annotation terms including both catabolism and nutrient transport may continue to accumulate beyond 72 hours.

## 1 Introduction

Charles Darwin described the Venus flytrap, *Dionaea muscipula* Ellis, as both "one of the most wonderful [plants] in the world" and a "horrid prison with closing walls" [1]. The Venus flytrap is perhaps best known for its active, snapping trap mechanism that it uses to capture prey for nutrition. This snap trap mechanism is shared with the related aquatic waterwheel plant, *Aldrovanda vesiculosa* L. (Droceraceae) [2], implying the origin of the snap trap in a common ancestor. Droseraceae and three other plant families with carnivorous species—

**Data Availability Statement:** Scripts and results for these analyses can be found at: https://github.com/mellamosummer/2022VenusFlyTrap/tree/main. Reads have been deposited in NCBI's SRA database under the project ID PRJNA981540.

**Funding:** This work was supported by the Francis Marion University professional development program (119145-100000-42105-82000), The University of Georgia, and the National Science Foundation (DEB-2110875). The funders had no role in study design, data collection and analysis, decision to publish, or preparation of the manuscript.

**Competing interests:** The authors have declared that no competing interests exist.

Drosophyllaceae, Nepenthaceae, Dioncophyllaceae—form a clade within the Caryophyllales along with the non-carnivorous family Ancistrocladaceae indicating an ancient origin of carnivory, and subsequent losses of carnivory in Ancistrocladaceae and two of three genera within Dioncophyllaceae [3, 4]. Carnivory has evolved independently more than five times in angiosperm history [5, 6] as an adaptation to thrive in nutrient-deficient soils in sunny, wet habitats [7]. The Venus flytrap is no exception to this rule, growing in semi-pocosins or semi-savannah areas [8].

The Venus flytrap closes in response to the stimulation of mechanosensing trigger hairs, which can be found on the adaxial side of the trap [1]. Stimulation of these hairs in rapid succession causes a very rapid shift in action potentials, two of which triggers trap closure [9–12]. Once a trap accumulates sufficient electrical charge from these action potentials, proton transport [13, 14], and ATP hydrolysis begin [15], aquaporin channels open [16], and water transport causes a rapid change in turgor across the trap [17]. Successful trap closure begins with increased trap turgor followed by a simultaneous expansion of the outer epidermis and a shrinkage of the inner epidermis [18, 19], changing the overall trap curvature from convex to concave, resulting in trap closure [20]. Upon properly sealing, a cocktail of digestive fluids is released into the interior of the trap and prey digestion takes place.

Over recent years, an array of digestive enzymes in the Venus flytrap have been well characterized. These digestive enzymes include a cysteine endopeptidase named dionain [21], as well as peroxidases, nucleases, phosphatases, phospholipases, a glucanase, chitinases, aspartic proteases, and a serine carboxypeptidase [22]. The initial release of digestive fluids is triggered by the action potentials generated from mechanical stimulation [23, 24]. These action potentials are stimulated until prey death and are important in the early establishment of the digestive cycle [25].

After prey death, the trap may require chemical feedback from partially digested prey to complete the release of digestive enzymes and promote nutrient absorption and transport. In aseptically grown *Drosera rotundifolia* L., for example, the application of crustacean chitin to the carnivorous plant leaves induced a marked increase in chitinase activity [26], demonstrating the link between enzyme release with the addition of chemical cues. Jakšová and others [27] showed that digestive enzyme regulation is not substrate specific, with bovine serum albumin (BSA) protein eliciting an upregulation of proteolytic enzymes, phosphatases, and chitinases.

At the same time, the trap absorbs nutrients–typically the domain of roots. After capture, the prey is digested, releasing amino acids and peptides. Glutamine is deaminated to produce ammonium ion ($NH_4^+$) which is then channeled into gland cells via the upregulated ammonium channel DmAMT1 [28]. Venus flytraps also use prey-derived amino acids as a substrate for cellular respiration [29] even with an abundance of atmospheric $CO_2$. In the carnivorous plant genus *Nepenthes*, transporters for amino acids and peptides are also expressed in the modified leaves [30].

Hormonal signaling also contributes to prey capture and digestion in carnivorous plants. The phytohormone jasmonic acid (JA) plays a critical role in both localized and systemic plant responses to injury (as reviewed in [31] through the synthesis of secondary metabolites [32] or by invoking changes in gene expression and growth form [33]. In *Lycoris aurea* (L'Hér.) Herb., for example, treatment of seedlings with methyl jasmonate resulted in 4,175 differentially expressed genes when compared to control plants [34]. Venus flytraps and other species in the carnivorous clade of the Caryophyllales also utilize the JA pathway to elicit prey capture and digestion responses [35–37].

Recently, a significant amount of work has gone into understanding the genomes of convergently evolved carnivorous plant lineages, including that of the Venus flytrap. The genome of

*Utricularia gibba* L., a carnivorous bladderwort, was found to have a haploid size of just 82 MBP/C in size while still encoding 28,500 genes [38], comparable to the number of genes in *Arabidopsis thaliana* [39]. *Genlisea aurea* A.St.-Hill, also in the Lentibulariaceae (Lamiales) is smaller still, with a nuclear genome size of only 64.6 MBP/C. These nuclear genomes are the two smallest vascular plant genomes sequenced to date.

Palfalvi and others [40] sequenced the genomes of three carnivorous plants in the Droseraceae (Caryophyllales), *Dionaea muscipula*, *Drosera spatulata* Labill., and *Aldrovanda vesiculosa* L. with genome sizes of 3,187 MBP/C, 293 MBP/C, and 509 MBP/C, respectively. The authors point to a genome duplication event in an ancestral Droseraceae species and an additional genome duplication event in the ancestor of *A. vesiculosa*. Venus flytrap has a much larger genome size than *Drosera* or *Aldrovanda* due to a massive expansion of long terminal repeat (LTR) retrotransposons, which comprise at least 38.78% of its genome. Palfalvi et al. [40] inferred a reduction in the number of protein-coding genes in the ancestral Droseraceae lineage, but interestingly, gene families with functions associated with prey attraction, nutrient uptake, and digestion increased in size. Here, we investigate change in transcript abundance over a time course following prey capture in Venus flytrap. We compare transcripts from traps that were triggered with and without prey and characterize sets of genes exhibiting similar temporal changes in transcript abundance over a 72-hour time course following prey capture. Our findings document a complex and dynamic transcriptional process with significant differences between triggered traps with and without prey seen in transcript profiles just one hour after closure.

## 2 Materials and methods

### 2.1 Plant care and total RNA isolation

Seed grown plants obtained from tissue culture stock provided by Michael Kane (University of Florida) were grown in an environmental chamber maintained at 22˚C supplemented with fluorescent lights which were cycled 16 hours on and 8 hours off a day. Plants were grown in a 1:1 mix of peat moss and perlite and watered with distilled water at under 50 parts per million of total dissolved solids. Experimental traps were fed with a single live darkling beetle (Tenebrionidae) larvae. To induce trap closure without a prey item introduced, a single trigger hair was stimulated two times in rapid succession with a metal probe. This action induces trap closure. Prior to RNA isolation, traps were harvested with sterile scissors just distal to the point of petiole attachment. Tissue was harvested from plants grown from seed that was derived from a single plant grown from tissue culture (tissue culture provided by Michael Kane, University of Florida). A single trap was harvested from each of the 44 plants randomly assigned to 'prey' and 'no prey' and time since trap closure treatments. Four replicates were sampled for each of 11 treatments: traps without prey harvested at 0 min, 5 min, 60 min, and 1440 min following mechanical triggering, and traps with prey were harvested at 5 min, 30 min, 60 min, 720 min, 1440 min, 2880 min, and 4320 min after triggering trap closure (Fig 1). Immediately prior to harvesting traps with prey, beetle larvae were cut longitudinally along one side and the prey item was extracted with sterile forceps. Traps were immediately placed into liquid nitrogen for tissue homogenization and RNA isolation. Additionally, we harvested and isolated total RNA from petiole tissue of the four traps at sampled immediately following mechanically stimulated closure (i.e. the 0 min, no prey treatment). Total RNA was isolated from each sample using the Direct-zol kit (Zymo Research, Irvine, CA, USA) with Plant RNA reagent (Life Technologies, Carlsbad, CA, USA). One 2880 min prey sample library failed leaving only three replicates for this treatment.

# Sampling Design

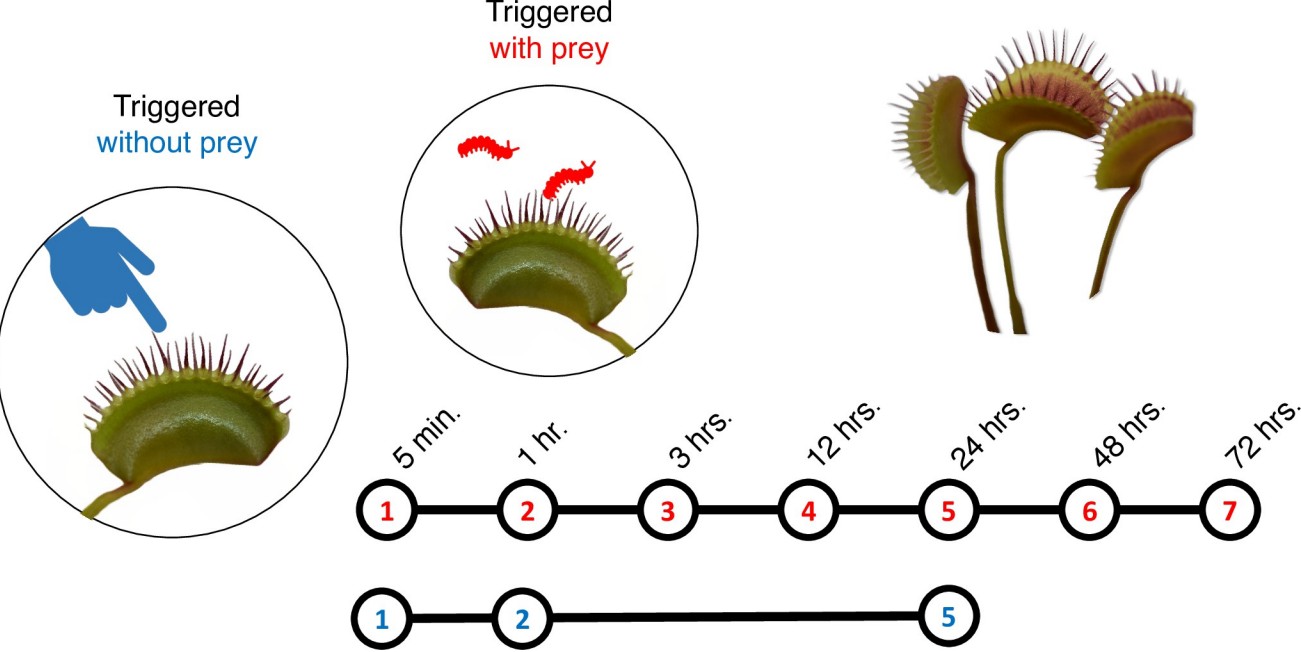

**Fig 1. A diagram of the sampling design.** Venus fly traps were either mechanically triggered (blue) or fed prey (red). Traps triggered with prey were harvested at 7 different time points between 5 minutes and 72 hours. Mechanically triggered traps were harvested at 3 different time points (5 minutes, 1 hour, and 24 hours). In addition, trap and petiole tissue was sampled from a set of control plants without triggering.

Bulk RNAs were isolated from each sampled trap with four biological replicates per treatment. Additionally, we isolated total RNA from petiole tissue and traps at 0 min. Total RNA was isolated using the Direct-zol kit (Zymo Research, Irvine, CA, USA) with Plant RNA reagent (Life Technologies, Carlsbad, CA, USA). One 2880 min prey sample library was dropped so we only had three replicates for these two treatments.

## 2.2 RNA-Seq and read processing

Illumina RNASeq library construction and RNA sequencing on an Illumina NovaSeq platform was conducted at HudsonAlpha Institute for Biotechnology. The median and minimum for read pairs after trimming are found in S1 Table. Trimmomatic v0.39 [41] was used for quality trimming and adapter clipping. Read quality was assessed using FASTQC v0.11.9 [42] & MultiQC v1.8 [43] before and after trimming. Pre- and post-trimming read counts are also reported in S1 Table).

## 2.3 Paired timepoint differential expression analysis

Trimmed reads were pseudo-aligned to the *D. muscipula* reference transcriptome index, then quantified using Kallisto v0.46.1 [44]. The transcriptome index was generated using published gene models and their annotations [40] downloaded from https://www.biozentrum.uni-wuerzburg.de/carnivorom/resources (*Dm_transcripts*).

A principal component analysis (PCA) was performed using the transcript abundance data with the R software package Sleuth [45] to visualize the variance between sample groups and

replicates. Transcript abundances were then used to conduct pair-wise differential gene expression analyses between triggered traps with and without prey sampled at 5 min, 1 h, & 24 h after triggering. The Benjamini-Hochberg approach [46] was used to account for multiple tests with a false discovery rate threshold of q < 0.05.

Differentially expressed transcripts were then translated using SeqKit [47] and protein sequences were BLASTed against annotated *Arabidopsis thaliana* proteins (Araport11_-pep_20220914) from The Arabidopsis Information Resource [48] and the Swiss-Prot database [49]. GO Term Enrichment was performed using the org.At.tair.db [50] and TopGO [51] packages in R.

## 2.4 Gene co-expression analysis

To characterize changes in transcript abundance across sampling time points, a gene coexpression analysis was performed on Transcripts per Million (TPM) data for both treatments and all time points. The analysis was performed in R using the Simple Tidy GeneCoEx workflow ([52] - https://github.com/cxli233/SimpleTidy_GeneCoEx) to identify sets of genes that display correlated expression patterns. Briefly, the workflow uses the Tidyverse package [53] to generate a gene correlation matrix including transcripts with >5 TPM in two or more samples. Transcripts were also filtered to remove genes exhibiting low variance in expression among prey treatments and time points (vartreatment logTPM < median vartreatment logTPM; F<2 for treatment effects) before testing for interactions. Genes with logTPM values exhibiting timepoint X treatment interactions with F> = 2 were used to generate a co-expression network and detect gene sets (modules) with >5 genes exhibiting highly correlated (r>0.75) transcript abundance profiles. The number of modules was optimized based on two indices: (1) the number of modules that have > = 5 genes and (2) the number of genes that are contained in modules that have > = 5 genes.

Annotations of genes within each module were inspected for terms related to catabolic processes including synthesis of digestive enzymes. GO Term Enrichment for each module was performed using the org.At.tair.db [54] and TopGO [55] packages in R.

## 3 Results

### 3.1 Paired time point differential expression analyses

Transcripts for 15146 and 14861 gene models were found 1 hour and 24 hours after triggering, respectively. We identified 174 and 151 differentially expressed genes between prey-fed and mechanically triggered ("no-prey") traps at 1 hour and 24 hour time points (q-val< = 0.05), respectively. Interestingly, no differentially expressed genes were identified between prey-fed and no-prey traps at the 5-minute time point.

At the 1-hour time point, 103 transcripts showed significantly higher expression in prey-fed traps, while 71 transcripts showed significantly higher expression in no-prey traps. Several GO terms were enriched among DE genes at this time point including: response to stimulus, regulation of jasmonic acid mediated signaling pathway, defense response to bacterium, response to salicylic acid, and photosynthesis (S1 Fig).

At the 24-hour time point, 149 transcripts showed significantly higher expression in prey-fed traps, while just 2 transcripts showed significantly higher abundance in no-prey traps. These include a gene homologous to the flowering-time gene CONSTANS and a basic leucine zipper transcription factor. Several GO terms were enriched at this time point including response to wounding, response to jasmonic acid, defense response to bacterium, response to salicylic acid, and transmembrane transport (S1 Fig).

## 3.2 Gene co-expression analysis

A Principal Component Analysis (PCA) was used to visualize variation among samples and treatments. (Fig 2). The prey treatments sampled at later time points (> 60 minutes) are well separated from the no-prey and earlier time points on PCA1 (Fig 2). Prey and no-prey treatments sampled at 60 minutes and earlier are not separated on PCA1. Whereas, comparisons of traps with and without prey did implicate differentially expressed genes at the 1- and 24-hour post-triggering timepoints (60 and 1440 min, respectively), the full transcript profiles for these two treatments are overlapping in the PCA plot (Fig 2).

Of the 14217 gene models for which abundance levels greater than five transcripts per million were observed after filtering, 10041 exhibited variation across time points. Transcript abundance profiles for these genes clustered into 14 coexpression modules, each with more than five genes (Fig 3A, S2 Fig). Modules were assigned numbered names by the GeneCoEx workflow, but as is evident in Fig 3, these names are not ordered with respect to module gene number or time signature.

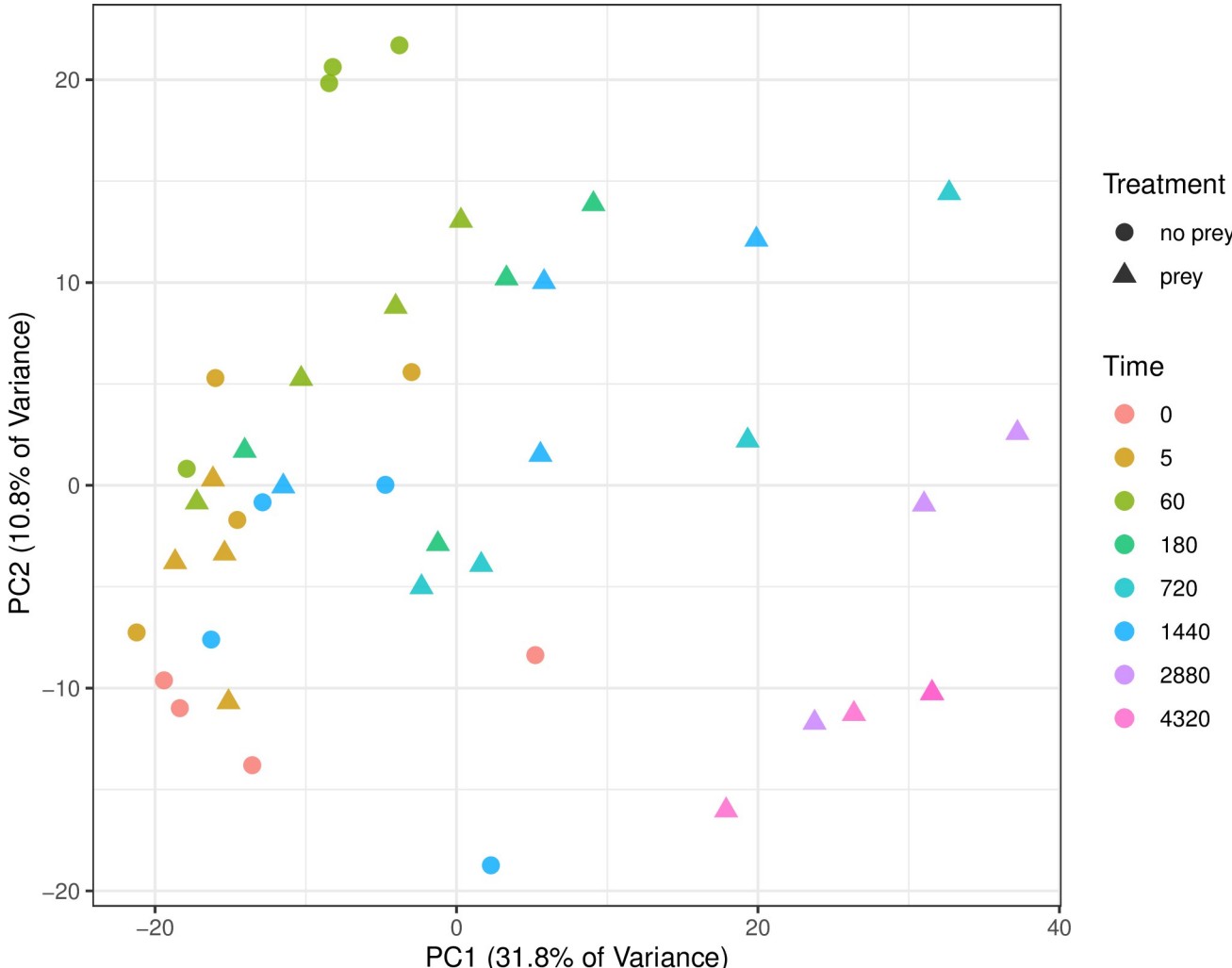

**Fig 2. Principal component analysis (PCA) of RNA-seq data from traps.** Replicates from the same time point are indicated with the same color while treatments are indicated by geometric shape (circle for mechanically triggered vs. triangle for prey-fed traps).

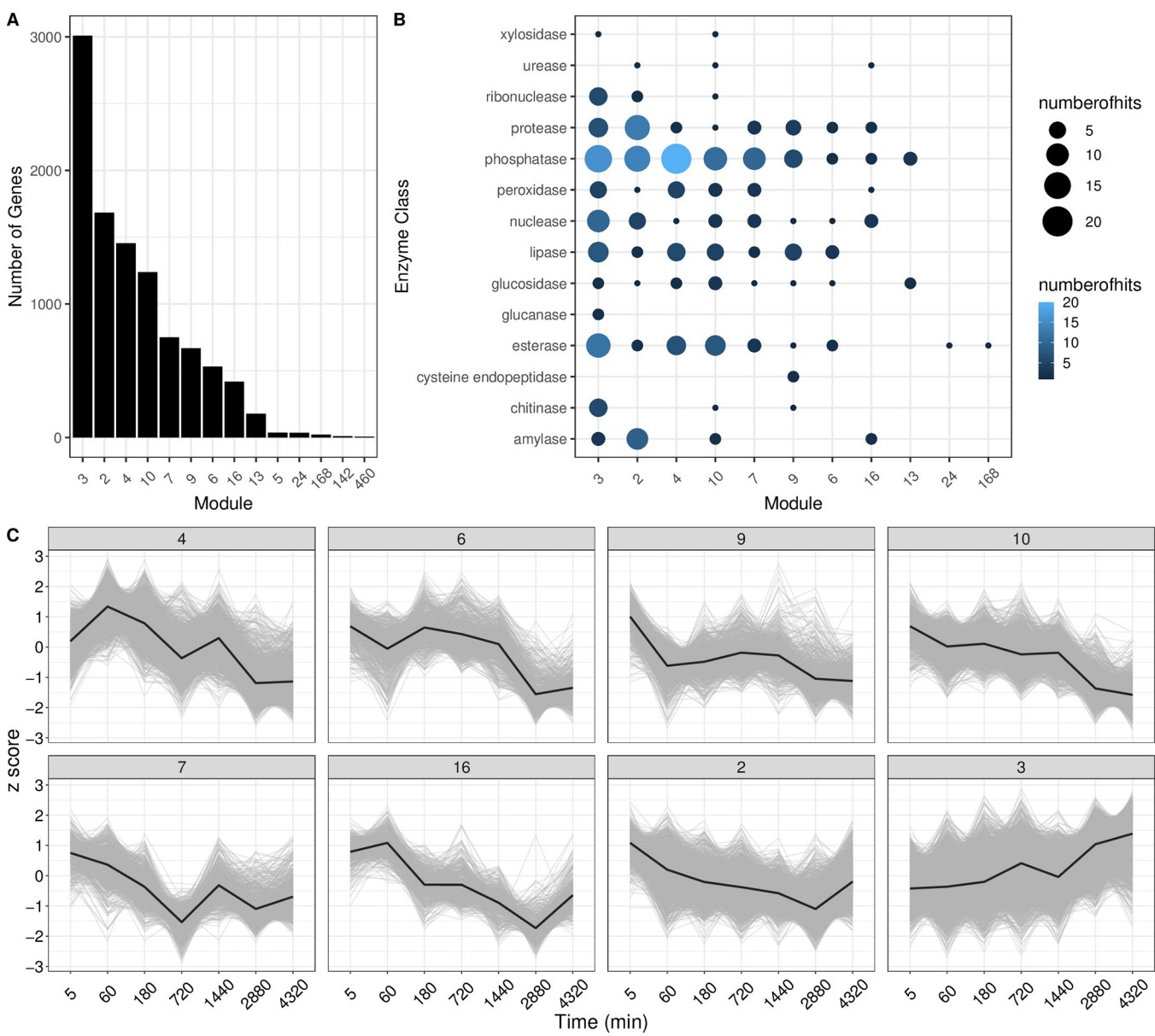

**Fig 3. Gene co-expression analysis.** A. Number of genes in each module with >5 highly co-expressed genes (n = 14). B. The number of BLAST hit descriptions for enzymes related to digestion are shown for each module. Color and size of each point depict the number of enzyme transcripts in each module. C. Expression Z-scores for each gene in modules with >400 genes (n = 8). Clusters represent co-expressed genes within each module. Mean expression is shown in the black line.

Focusing on the eight modules with more than 400 genes, Fig 3C shows that most module profiles exhibit peak transcript between 5 and 60 minutes after prey capture. Interestingly, the largest module (n = 3009, Fig 3A, 3B module "3"; Fig 3C bottom right panel), including ~21% of transcripts with time-structured expression, is the exception with genes showing increasing transcript abundance from 5 minutes until the last time point at 72 hours after prey capture.

Several classes of enzymes were represented in each module (Fig 3B). Phosphatases were most abundant across modules. Cysteine endopeptidases were only found in module 9. Glucanases were only found in module 3 which exhibited peak expression at 72hrs. Additionally, many genes from the prey/no-prey DE gene lists were found among the time-signature

modules. Module 3 contained 29 and 83 genes from the 1hr and 24hrs, prey/no-prey differential expression analysis, respectively. Module 4 contained 57 and 64 genes from the 1hr and 24 pair-wise differential expression analyses, respectively. Module 10 contained 35 genes from the 1hr pair-wise differential expression analysis. (S3 Fig)

## 4 Discussion

### 4.1 Transcriptomic differences between traps with prey items and 'no-prey' traps

It is thought that the relative rarity of botanical carnivory is, at least in part, due to the narrow range of conditions under which the habit is advantageous from a cost/benefit perspective [7]. Thus, it would be reasonable to assume that the overall fitness of the flytrap is at least partially dependent on the ability of the plant to detect failed prey capture attempts and ultimately reset the trap. It is well known that the Venus flytrap plants respond differently to triggering of traps with and without prey. The generation of the action potentials driving triggered trap closure results in a temporary decrease in photosynthetic activity to nearly zero. In traps retaining prey, the decrease in photosynthetic activity is prolonged for as long as the trap is receiving mechanical stimulation [56]. Therefore, triggered traps without prey temporarily lose the ability to capture new prey and experience a decrease in their photosynthetic rate without increased nutrient absorption from prey being digested.

The resetting of a falsely triggered trap is a relatively long process–on the order of several days [57] and there are outstanding questions about what is happening at the transcriptomic level during much of this time. To that end, we examined the full transcriptomes of traps triggered with and without prey items at 5 min, 60 min, and 1440 min (24 hrs.). Interestingly, at the five-minute time point we detect no genes being differentially expressed between traps with and without prey, suggesting the trap has not yet determined, at the transcriptomic level, whether or not it has captured a suitable prey item. Of course, a lack of activity at the scale of the transcriptome does not indicate that no differences exist between treatments. In fact, it is known that, in closed traps containing prey, the action potentials generated by struggling insects disturbing trigger hairs cause a spike in cytosolic calcium levels associated with downstream Jasomonic Acid signaling [58].

At the 60-minute time point, we identified 174 significantly differentially expressed genes between prey-fed and no-prey traps. Of the 174 DEGs, 103 genes showed higher expression in traps containing prey and 71 genes showed higher expression in traps without prey. By 60 minutes after trap closure, the annotations of transcripts exhibiting significantly higher abundance with vs. without prey imply 1) the activation of the jasmonic acid pathway and other responses to wounding, 2) a general response to bacterial and fungal pathogens, and 3) nutrient absorption. Pathogenesis-related (PR) proteins have been described in the pitcher fluid of the carnivorous *Nepenthes alata*. The putative function of these proteins is to suppress the proliferation of putrefying bacteria on captured prey items or in the pitcher fluid [59]. Unlike digestion in *Dionaea* traps, *Nepenthes* pitchers do not enclose their prey after capture, and pitchers are known to host a diverse microbiome community in their digestive fluid [60]. For this reason, it may be somewhat surprising to see PR-related genes being upregulated in *Dionaea*; however, despite the sealed environment of the *Dionaea* trap, prey items may well harbor putrefying bacteria or fungi. PR proteins have been detected in the secretome of *Dionaea* previously [22], however, the detection of PR gene transcripts after only an hour may suggest a primed response to resist putrifying bacteria introduced by prey who might complete for liberated nutrients [61]. Simultaneously, the upregulation of certain nutrient transport genes (*e.g.* AKINBETA1, atLHT1) at this time implies that the trap is primed for nutrient absorption only one hour after prey capture, possibly before the secretion of

digestive fluid into the lumen of the closed trap. In contrast to traps with prey, those without prey exhibit an increased abundance of transcripts for proteins to be transported to chloroplasts, perhaps contributing to the repair of chlorophyll or restoration of photosynthesis prior to the trap resetting. Reactive oxygen species are known to spike during prey capture and digestion [62] and evidence from other sources of oxidative stress suggests that these molecules can be particularly damaging to chlorophyll [63, 64], so it would not be surprising if the stress of trap closure damaged chlorophyll via the generation of reactive oxygen species.

We identified 151 significantly differentially expressed genes between prey-fed and no-prey traps at 1440 m (24 hrs.). Of the 151 DEGs, 149 genes showed higher expression in traps containing prey, while only 2 genes showed higher expression in traps without prey. Again, the 1440-minute time point is rich in genes related to plant defense and the jasmonic acid pathway. Additionally, genes related to nutrient transport continue to exhibit higher abundance in traps with prey. Unlike the one-hour time point, at 24 hours, transcripts coding for enzymes directly related to digestion have significantly higher abundance in traps with prey: three serine-type carboxypeptidases (AT3G10410, AT1G15000, AT2G27920) two aspartic-type endopeptidases (AT3G25700 and AT2G03200), and a cysteine endopeptidase (AT5G45890), all known to be associated with the *Dionaea* secretome [22]. Taken together, we can infer based on the transcriptomic data presented here that transcriptional initiation of the digestion process does not occur before five minutes post prey capture. By 60 minutes, however, traps with prey are primed for digestion and nutrient absorption, and traps without prey have perhaps begun to repair damaged tissue ahead of trap resetting, based on the high level of localization to the chloroplast. By 24 hours, the traps with prey have also begun to upregulate enzyme-coding transcripts related to catabolism in addition to the continuation of defense response, response to pathogens, and the production of nutrient transporters. At the same time, traps triggered without prey show the differential expression of very few genes although the trap remains closed–suggesting a refractory period before the trap resetting.

## 4.2 Time series analysis

Generally speaking, our time structured experiment reveals that the process of prey capture, digestion, and nutrient absorption is a dynamic and complex process involving the transcriptional orchestration of roughly 50% of protein-coding genes in *Dionaea muscipula*. Module 3 is the most gene rich module (n = 3009) and is the only module whose peak expression is seen at the end of our time series (72 hours). This module also contains an abundance of enzyme-encoding genes with annotations related to prey digestion (Fig 3B), suggesting that the digestive process remains dynamic even after several days of activity. In fact, this late peaking module is enriched in chitinase activity, amylase activity, and aspartic endopeptidase activity when compared to the modules with peak expression within a few hours of prey capture (Fig 3C). The enrichment of transcripts for some of these enzyme classes directly related to digestion suggest that this peak of expression isn't simply related to the consequences of digestion (e.g. cell repair or senescence), but may be indicating a changing cocktail of trap fluid proteins with prey digestion and nutrient absorption functions. Another possibility is that the concentrations of digestive enzymes peak earlier than 72 hours, but the transcripts included in module 3 continue to accumulate. Further work on the dynamics of prey digestion and nutrient absorption in traps should integrate proteomic and metabolomic data.

## 4.3 Conclusion

Comparisons of transcript profiles for triggered *Dionaea muscipula* traps with and without prey suggest that plants decide somewhere between the five- and sixty-minute timepoints

whether they have captured suitable prey and alter their transcriptome accordingly. Differences in the transcription profiles of traps with and without prey emerge well before the trap hermetically seals and, of course, before the secretion of digestive enzymes. The upregulation of several key genes related to nutrient absorption and transport occurred early and preceded the upregulation of some of the characteristic digestive enzymes, such as the various endopeptidases. This observation suggests that prey-derived macromolecules do not need to be liberated to initiate the absorptive function of traps. Many transcripts show peak abundance within an hour of prey capture, but more than 3000 genes exhibit increasing transcript abundance up to and likely beyond 72 hours post-capture including an enrichment of amylase activity, aspartic endopeptidase activity, and chitinase activity. This observation suggests a dynamic and robust transcriptomic response related to prey digestion and nutrient absorption that continues up to and likely after three days after prey capture. In addition to integrating transcriptomic, proteomic and metabolomic analyses, better understanding of Venus flytrap prey digestions and nutrient absorption dynamics could come with increased timepoints in a time series analysis, extending beyond 72 hours.

## Supporting information

**S1 Fig.** Gene ontology dotplots including: (1) Top 30 Enriched Gene Ontology Biological Process Terms for differentially expressed genes between prey/no prey traps at 1 hr. The Rich Factor is the ratio of the number of enriched DEGs in the GO BP category. Point size shows the number of genes assigned to each GO category. The -log10(P value) is represented by a color scale. (2) Top 30 Enriched Gene Ontology Molecular Function Terms for differentially expressed genes between prey/no prey traps at 1 hr. The Rich Factor is the ratio of the number of enriched DEGs in the GO MF category. Point size shows the number of genes assigned to each GO category. The -log10(P value) is represented by a color scale. (3) Top 19 Enriched Gene Ontology Cellular Component for differentially expressed genes between prey/no prey traps at 1 hr. The Rich Factor is the ratio of the number of enriched DEGs in the GO CC category. Point size shows the number of genes assigned to each GO category. The -log10(P value) is represented by a color scale. (4) Top 30 Enriched Gene Ontology Biological Process Terms for differentially expressed genes between prey/no prey traps at 24 hr. The Rich Factor is the ratio of the number of enriched DEGs in the GO BP category. Point size shows the number of genes assigned to each GO category. The -log10(P value) is represented by a color scale. (5) Top 30 Enriched Gene Ontology Molecular Function Terms for differentially expressed genes between prey/no prey traps at 24 hr. The Rich Factor is the ratio of the number of enriched DEGs in the GO MF category. Point size shows the number of genes assigned to each GO category. The -log10(P value) is represented by a color scale. (6) Top 23 Enriched Gene Ontology Cellular Component for differentially expressed genes between prey/no prey traps at 1 hr. The Rich Factor is the ratio of the number of enriched DEGs in the GO CC category. Point size shows the number of genes assigned to each GO category. The -log10(P value) is represented by a color scale.
(PDF)

**S2 Fig. Expression Z-scores for each gene in each of the modules with >5 genes (n = 14).** Clusters represent co-expressed genes within each module by treatment (prey vs. no prey). Mean expression is shown in the black line.
(PDF)

**S3 Fig.** Number of differentially expressed genes in each co-expression module, including (1) differentially expressed genes between traps triggered with and without prey at 1 hr and 24 hr

time points. Color shows which treatment had statistically higher expression. (2) The number of differentially expressed genes at the 1 hour time point found amongst the top 8 Gene Co-Expression modules. Color shows which treatment had statistically higher expression. (3) The number of differentially expressed genes at the 24 hour time point found amongst the top 8 Gene Co-Expression modules. Color shows which treatment had statistically higher expression.
(PDF)

**S1 Table. Pre- and post-trimming read counts.**
(XLSX)

## Acknowledgments

We thank Ingrid Jordan Thaden for her thoughtful advice on RNA isolation techniques and Michael Kane for *Dionaea muscipula* tissue culture. We thank Jane Grimwood, Director of the Genome Sequencing Center (GSC) at HudsonAlpha Institute for Biotechnology, for providing outstanding sequencing services. We also thank the systems administrators and support staff at the Georgia Advanced Computing Resource Center (GACRC) for maintaining the high-performance computing platforms used for all our bioinformatic analyses.

## Author Contributions

**Conceptualization:** Jeremy D. Rentsch, James H. Leebens-Mack.

**Data curation:** Summer Rose Blanco.

**Formal analysis:** Summer Rose Blanco, James H. Leebens-Mack.

**Methodology:** Jeremy D. Rentsch.

**Project administration:** Jeremy D. Rentsch.

**Resources:** Jeremy D. Rentsch.

**Supervision:** Jeremy D. Rentsch.

**Writing – original draft:** Jeremy D. Rentsch, Summer Rose Blanco, James H. Leebens-Mack.

**Writing – review & editing:** Jeremy D. Rentsch, James H. Leebens-Mack.

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
