## [Decision Letter · Decision Letter 0]

6 Feb 2024

PONE-D-23-37511Comparative transcriptomics of Venus flytrap (Dionaea muscipula) across stages of prey capture and digestion.PLOS ONE

Dear Dr. Rentsch,

Thank you for submitting your manuscript to PLOS ONE. After careful consideration, we feel that it has merit but does not fully meet PLOS ONE’s publication criteria as it currently stands. Therefore, we invite you to submit a revised version of the manuscript that addresses the points raised during the review process.

**As you can see in attached reports, both of the reviewers concern the M & M. Please provide more information about M&M. Moreover, please improve the manuscript according to the reviewers' suggestions.**

We look forward to receiving your revised manuscript.

Kind regards,

Yonggen Lou

Academic Editor

PLOS ONE

 [This work was supported by the Francis Marion University professional development program (119145-100000-42105-82000), The University of Georgia, and the National Science Foundation (DEB-2110875).].  

4. We note that Figure 1 in your submission contain copyrighted images. All PLOS content is published under the Creative Commons Attribution License (CC BY 4.0), which means that the manuscript, images, and Supporting Information files will be freely available online, and any third party is permitted to access, download, copy, distribute, and use these materials in any way, even commercially, with proper attribution. For more information, see our copyright guidelines: http://journals.plos.org/plosone/s/licenses-and-copyright.

Reviewers' comments:

Reviewer's Responses to Questions

**Comments to the Author**

1. Is the manuscript technically sound, and do the data support the conclusions?

Reviewer #1: Yes

Reviewer #2: Yes

2. Has the statistical analysis been performed appropriately and rigorously? 

Reviewer #1: Yes

Reviewer #2: No

3. Have the authors made all data underlying the findings in their manuscript fully available?

Reviewer #1: Yes

Reviewer #2: Yes

4. Is the manuscript presented in an intelligible fashion and written in standard English?

Reviewer #1: Yes

Reviewer #2: No

5. Review Comments to the Author

Reviewer #1: The manuscript by Rentsch et al. reports on the transcriptional responses of Venus flytrap (Dionaea muscipula) to either mechanical stimulation or by prey ingestion at various time points. By comparing the transcriptome differences at different times, the authors identified potential prey digestion- and nutrient absorption-related genes. Employing co-expression analysis, the authors clustered genes into different modules.

1. MM should be written in more detail. How was mechanical stimulation was done? I could not find any description of it in MM or Results. How many larvae were given to each trap? How many biological replicates were harvested for each sample group?

2. I would suggest downsizing the Discussion section. May part of Discussion which seems to be over-discussed. For example, L262-268, while AtNPR1 is indeed a crucial regulator of SA signaling, its presence does not necessarily indicate the involvement of the entire JA and SA pathway. Other evidence is needed to support that “Perhaps there is a function here in the moderation between the defense response and the wounding response”.

3. Generally specific data, including genes and pathways, should not appear only in Discussion, e,g., AtNPR1 and AKINBETA1. These should be given in the Results (, before being discussed.

4. L242-245. Although no DEGs were detected, certain rapid responses, such as calcium signaling or MAPK signaling, may have bee activated. The trap might have perceived the triggering stimulations already, just not on the transcriptional level.

5. L261, please elaborate “suggest a dual role in both pathogen defense and prey digestion”. Why do PR proteins have functions in prey digestion?

6. Caryophyllales already possesses ample genomic data. The authors could analyze the evolutionary history of identified candidate genes, exploring carnivory in Caryophyllales from an evolutionary perspective.

7. Please use “h” for “hour” and use “min”, but not “m” (m = meter, not minute).

Reviewer #2: In this manuscript, authors compared transcriptional differences between venus flytrap with vs without prey in time series. The major achievement is that venus flytrap recognizes a successful trap after 5 min, probably starts at 1 hour, and the catabolism and nutrient transport will continue to beyond 72 hours. Briefly, authors preliminarily described the transcriptomic at three time points, without deep analysis and conclusive discovery.

Major comments:

1, The description to methods is rough. How traps without prey are treated? “mechanical triggering” (L122) is not clear enough to understand whether this is a perfect control for traps with prey. Is there mechanical damage or only touch?

2, for transcriptomic analysis, DEG analysis must be adjusted. In time series analysis, about 50% protein-coding genes are differently expressed. How this DEG were adjusted. There is no statistic analysis part in Methods.

3, L19-194 said that “the first principal component appears to correlate with the time”. No clear results support this interpretation.

4. The first paragraph of 3.2 (L191-197) cited Figure 2 rather than Figure 1. However, there is only one panel in Figure 2, rather than Figure 1a and Figure 1b. Please carefully check Figure 2.

5, Does the “highly coexpressed genes” (L200) mean that gene abundance is greater than five transcripts per million? Please clarify.

The font size of axis annotations in Figure 3a and 3b is too small to read.

6. PLOS authors have the option to publish the peer review history of their article (what does this mean?). If published, this will include your full peer review and any attached files.

Reviewer #1: No

Reviewer #2: No

---

## [Author Response · Author response to Decision Letter 0]

26 Apr 2024

 Response: Okay great. 

Response: This has been addressed and updated in the document. 

 [This work was supported by the Francis Marion University professional development program (119145-100000-42105-82000), The University of Georgia, and the National Science Foundation (DEB-2110875).]. 

Response: This has been addressed and updated in the document. 

4. We note that Figure 1 in your submission contain copyrighted images. All PLOS content is published under the Creative Commons Attribution License (CC BY 4.0), which means that the manuscript, images, and Supporting Information files will be freely available online, and any third party is permitted to access, download, copy, distribute, and use these materials in any way, even commercially, with proper attribution. For more information, see our copyright guidelines: http://journals.plos.org/plosone/s/licenses-and-copyright.

Response: We believe you are referring to the pictures of the flytraps themselves. Those are photographs taken by first author, Jeremy D. Rentsch, who gives permission for their use. If this isn’t specifically what is meant please clarify and be specific. 

2. Has the statistical analysis been performed appropriately and rigorously?

Reviewer #1: Yes

Reviewer #2: No

Response: We believe the updates to the methodology should satisfy this requirement for reviewer 2. 

4. Is the manuscript presented in an intelligible fashion and written in standard English?

Reviewer #1: Yes

Reviewer #2: No

Response: No specifics provided and we found no such issues to correct in the manuscript. 

5. Review Comments to the Author

Reviewer #1: The manuscript by Rentsch et al. reports on the transcriptional responses of Venus flytrap (Dionaea muscipula) to either mechanical stimulation or by prey ingestion at various time points. By comparing the transcriptome differences at different times, the authors identified potential prey digestion- and nutrient absorption-related genes. Employing co-expression analysis, the authors clustered genes into different modules.

1. MM should be written in more detail. How was mechanical stimulation was done? I could not find any description of it in MM or Results. How many larvae were given to each trap? How many biological replicates were harvested for each sample group?

Response: We have added additional details to the methodology that should remedy each of these questions. 

2. I would suggest downsizing the Discussion section. May part of Discussion which seems to be over-discussed. For example, L262-268, while AtNPR1 is indeed a crucial regulator of SA signaling, its presence does not necessarily indicate the involvement of the entire JA and SA pathway. Other evidence is needed to support that “Perhaps there is a function here in the moderation between the defense response and the wounding response”.

Response: The discussion has been downsized significantly and the comment you reference here has been removed for being superfluous. 

3. Generally specific data, including genes and pathways, should not appear only in Discussion, e,g., AtNPR1 and AKINBETA1. These should be given in the Results (, before being discussed.

4. L242-245. Although no DEGs were detected, certain rapid responses, such as calcium signaling or MAPK signaling, may have bee activated. The trap might have perceived the triggering stimulations already, just not on the transcriptional level.

 Response: We’ve added clarity to this remark – making clear we are only speaking at the transcriptomic level and then reinforcing the idea by discussing some signaling that occurs very early. 

5. L261, please elaborate “suggest a dual role in both pathogen defense and prey digestion”. Why do PR proteins have functions in prey digestion?

Response: Clarity has been brought to this portion. Many enzymes related to digestion are in PR gene families, but I failed to make a meaningful connection here and made a more direct connection instead. 

6. Caryophyllales already possesses ample genomic data. The authors could analyze the evolutionary history of identified candidate genes, exploring carnivory in Caryophyllales from an evolutionary perspective.

7. Please use “h” for “hour” and use “min”, but not “m” (m = meter, not minute).

Response: This has been fixed throughout. Thank you. 

Reviewer #2: In this manuscript, authors compared transcriptional differences between venus flytrap with vs without prey in time series. The major achievement is that venus flytrap recognizes a successful trap after 5 min, probably starts at 1 hour, and the catabolism and nutrient transport will continue to beyond 72 hours. Briefly, authors preliminarily described the transcriptomic at three time points, without deep analysis and conclusive discovery.

Response: Thank you for your time as a reviewer. We attempted to clarify and tighten up the manuscript to highlight a few of our exciting finds. We are excited to see what work comes of this. 

Major comments:

1, The description to methods is rough. How traps without prey are treated? “mechanical triggering” (L122) is not clear enough to understand whether this is a perfect control for traps with prey. Is there mechanical damage or only touch?

Response: We have added some additional clarity to the methodology. This includes a more specific description of trap stimulation with a metal probe for the ‘non-prey’ treatment. 

2, for transcriptomic analysis, DEG analysis must be adjusted. In time series analysis, about 50% protein-coding genes are differently expressed. How this DEG were adjusted. There is no statistic analysis part in Methods.

 Response: Sleuth uses the Benjamini-Hochberg approach (Benjamini & Hochberg, 1995). We added a sentence and citation of B&H to the methods for clarity. 

3, L19-194 said that “the first principal component appears to correlate with the time”. No clear results support this interpretation.

Response: We clarified the language in around this topic. 

4. The first paragraph of 3.2 (L191-197) cited Figure 2 rather than Figure 1. However, there is only one panel in Figure 2, rather than Figure 1a and Figure 1b. Please carefully check Figure 2.

Response: Thank you for your attention on this. We fixed our error. 

5, Does the “highly coexpressed genes” (L200) mean that gene abundance is greater than five transcripts per million? Please clarify.

Response: We clarified the language in this matter to be less ambiguous (e.g. ( Transcript abundance profiles for these genes clustered into 14 coexpression modules, each with more than five genes)

The font size of axis annotations in Figure 3a and 3b is too small to read.

Response: We increased the font size and believe the figu

---

## [Decision Letter · Decision Letter 1]

24 May 2024

Comparative transcriptomics of Venus flytrap (Dionaea muscipula) across stages of prey capture and digestion.

PONE-D-23-37511R1

Dear Dr. Rentsch,

We’re pleased to inform you that your manuscript has been judged scientifically suitable for publication and will be formally accepted for publication once it meets all outstanding technical requirements.

Kind regards,

Yonggen Lou

Academic Editor

PLOS ONE

Additional Editor Comments (optional):

Reviewers' comments:

Reviewer's Responses to Questions

**Comments to the Author**

1. If the authors have adequately addressed your comments raised in a previous round of review and you feel that this manuscript is now acceptable for publication, you may indicate that here to bypass the “Comments to the Author” section, enter your conflict of interest statement in the “Confidential to Editor” section, and submit your "Accept" recommendation.

Reviewer #1: All comments have been addressed

2. Is the manuscript technically sound, and do the data support the conclusions?

Reviewer #1: (No Response)

3. Has the statistical analysis been performed appropriately and rigorously? 

Reviewer #1: (No Response)

4. Have the authors made all data underlying the findings in their manuscript fully available?

Reviewer #1: (No Response)

5. Is the manuscript presented in an intelligible fashion and written in standard English?

Reviewer #1: (No Response)

6. Review Comments to the Author

Reviewer #1: (No Response)

7. PLOS authors have the option to publish the peer review history of their article (what does this mean?). If published, this will include your full peer review and any attached files.

Reviewer #1: No

---

## [Editor Report · Acceptance letter]

3 Jul 2024

PONE-D-23-37511R1 

PLOS ONE

Dear Dr. Rentsch, 

I'm pleased to inform you that your manuscript has been deemed suitable for publication in PLOS ONE. Congratulations! Your manuscript is now being handed over to our production team.

Kind regards, 

on behalf of

Dr. Yonggen Lou 

Academic Editor

PLOS ONE